# Intensity Differences of Resistance Training for Type 2 Diabetic Patients: A Systematic Review and Meta-Analysis

**DOI:** 10.3390/healthcare11030440

**Published:** 2023-02-03

**Authors:** Tenglong Fan, Man-Hsu Lin, Kijin Kim

**Affiliations:** 1Department of Physical Education, Keimyung University, Daegu 42601, Republic of Korea; 2Department of Sport Marketing, Keimyung University, Daegu 42601, Republic of Korea

**Keywords:** resistance training, type 2 diabetes, blood glucose, lipids

## Abstract

Resistance training is used as adjunctive therapy for type 2 diabetes (T2DM), and the aim of this study was to investigate the differences in the treatment effects of different intensities of resistance training in terms of glycemia, lipids, blood pressure, adaptations, and body measurements. A comprehensive search was conducted in the PubMed, EMBASE (Excerpta Medica dataBASE), EBSCO (Elton B. Stephens Company) host, Cochrane Library, WOS (Web of Science), and Scopus databases with a cut-off date of April 2022, and reference lists of relevant reviews were also consulted. The literature screening and data extraction were performed independently by two researchers. RoB2 (Risk of bias 2) tools were used for the literature quality assessment, the exercise intensity was categorized as medium-low intensity and high intensity, and the meta subgroup analysis was performed using R Version. A fixed or random effects model was selected for within-group analysis based on the heterogeneity test, and a random effects model was used for the analysis of differences between subgroups. A total of 36 randomized controlled trials were included, with a total of 1491 participants. It was found that resistance training significantly improved HbA1c (glycated hemoglobin), fasting blood glucose, TG (triglycerides), TC (total cholesterol), and LDL (low-density lipoprotein cholesterol) levels in patients with T2DM and caused a significant reduction in systolic blood pressure, percent of fat mass, and HOMA-IR (homeostatic model assessment for insulin resistance) indexes. The effects of high and medium-low intensity resistance training on T2DM patients were different in terms of HOMA-IR, maximal oxygen consumption, weight, waist-to-hip ratio, and body mass indexes. Only medium-low intensity resistance training resulted in a decrease in HOMA-IR. In addition to weight (MD = 4.25, 95% CI: [0.27, 8.22], *I*^2^ = 0%, *p* = 0.04; MD = −0.33, 95% CI: [−2.05, 1.39], *I*^2^ = 0%, *p* = 0.76; between groups *p* = 0.03) and HOMA-IR (MD = 0.11, 95% CI: [−0.40, −0.63], *I*^2^ = 0%, *p* = 0.85; MD = −1.09, 95% CI: [−1.83, −0.36], *I*^2^ = 87%, *p* = < 0.01; between groups *p* = 0.0085), other indicators did not reach statistical significance in the level of difference within the two subgroups of high intensity and medium-low intensity. The treatment effects (merger effect values) of high intensity resistance training were superior to those of medium-low intensity resistance training in terms of HbA1c, TG, TC, LDL levels and diastolic blood pressure, resting heart rate, waist circumference, fat mass, and percentage of fat mass. Therefore, high intensity resistance training can be considered to be a better option to assist in the treatment of T2DM and reduce the risk of diabetic complications compared to medium-low intensity resistance training. Only one study reported an adverse event (skeletal muscle injury) associated with resistance training. Although results reflecting the difference in treatment effect between intensity levels reached no statistical significance, the practical importance of the study cannot be ignored.

## 1. Introduction

Previous studies have shown that mortality from diabetes is declining in high-income countries; however, due to differences in demographics and lifestyles, the impact of diabetes on society is still expected to increase, especially in developing countries [1]. Additionally, the coronavirus pandemic has been ongoing for more than two years; until 28 June 2022, cases have found from more than 200 countries, more than 540 million people have been found to be infected, and more than 6.3 million have died [2]. A survey in the United States showed that diabetic comorbidities patients accounted for one-third of hospitalized patients who were infected by coronavirus [3]. A survey in China showed that diabetic comorbidities patients accounted for 19% of hospitalized coronavirus-infected patients [4]. Clinical investigations have shown that patients with type 2 diabetes and diabetic comorbidities are at high risk of admission to intensive care units and death after infection [5,6]. Thus, the impacts of type 2 diabetes on human health in the post-pandemic era cannot be ignored. 

Exercise can be an adjunctive therapy of type 2 diabetes, and its effectiveness in improving blood glucose, blood lipids, and other physiological indicators has been demonstrated [7,8,9]. A meta-analysis demonstrated significant differences in glycemic and insulin reductions between different intensity subgroups in patients with type 2 diabetes after a period of resistance training [10]. Ishiguro et al. [11] also found that resistance training can be recommended to patients who are in the early stage of type 2 diabetes for glycemic control and that patients with lower levels of obesity gained more benefits in this process. 

However, thanks to these two meta-analysis studies, we found that: (i) the included literature differed largely in these two studies; (ii) only glycemic indicators was included in these two meta-analyses, while lipid indicators, obesity level, and blood pressure were not studied. The blood lipid indicators total cholesterol, HDL (high-intensity lipoprotein) cholesterol, and LDL cholesterol, offer better prediction on atherosclerotic cardiovascular disease comorbidity than glycated hemoglobin [12,13]. Moreover, since obese people are considered to be a high-risk group in coronavirus-infected patients, indicators of blood lipids and obesity (i.e., body weight, BMI) need to be studied more thoroughly. Even though Yang et al. [14] found statistically significant effects in terms of resistance training on glycated hemoglobin, body mass index, peak oxygen consumption, and maximal heart rate in a group of patients with type 2 diabetes, research related to the therapeutic effects of different levels of intensity on various physiological indicators of type 2 diabetic patients was not conducted.

In general, in the previous meta-analysis studies considered resistance training as an effective treatment for type 2 diabetes patients, there was a lack of study of the intensity of exercise and the effectiveness to those indicators mentioned above. The criteria for selecting an optimum exercise intensity in the process of performing resistance training as an effective treatment for type 2 diabetes patients still needs to be discovered. The purpose of this study was therefore to understand the effectiveness of exercise at different intensity levels in terms of the critical indicators for type 2 diabetes patients.

## 2. Materials and Methods

### 2.1. Data Sources and Retrieval Strategies

The PRISMA statement guided the study, which involved searching the PubMed, EMBASE, EBSCO host, Cochrane Library, WOS, and Scopus databases with a search deadline of 5 April 2022. The search strategy used a combination of medicine subject heading terms and test words provided by the databases (see Appendix B: Search strategies). The literature retrieved from the six databases was downloaded and imported into Endnote x9, and the overlapped studies were automatically and manually deleted in it. Two personnel sorted the remaining studies into five categories, which were (i) not strongly associated with resistance training and type 2 diabetes, (ii) interventions disqualified, (iii) no randomized controlled trials conducted, (iv) experimental subjects not type 2 diabetic patients, and (v) full text (including conference literature) not available. This study was registered with PROSPERO (International Prospective Register of Systematic Reviews), under registration number: CRD42022326530.

### 2.2. Inclusion and Exclusion Criteria

#### 2.2.1. Inclusion Criteria

The included studies (i) were randomized controlled trials conducted, written in English, (ii) had type 2 diabetic patients aged ≥17 years as subjects, (iii) arranged only resistance training intervention to the experimental group, no exercise or only stretching to the control group, and (iv) included at least one indicators below in the results: glycated hemoglobin, insulin, HOMA (homeostasis model assessment), fasting glucose, TG (triglyceride), TC (total cholesterol), HDL, LDL, systolic blood pressure, diastolic blood pressure, BMI, body weight, waist circumference, and waist-hip ratio.

#### 2.2.2. Exclusion Criteria

Studies were excluded which (i) conducted randomized controlled trials with an intervention period of <4 weeks, (ii) included non-type 2 diabetic patients as subjects, (iii) obtained no mean ± standard deviation at experimental result data, and (iv) replicated experimental results in publications.

### 2.3. Data Extraction and Literature Quality Assessment

Two researchers used Microsoft Excel to extract data from the included studies and organized them using same sample tables. According to the collected results, the abnormal data were reviewed, and the difference from two tables was verified by rechecking the studies. The extracted data were as followed: (i) basic information from the study (e.g., author, country, year of publication, the number of intervention and control groups, and the characteristics of the participating population, etc.), (ii) resistance training plan (i.e., training duration, training intensity, and training frequency, etc.), and (iii) randomized controlled trials-related physiological indicators.

RoB2 tools were used to evaluate the quality of the literature according to five aspects, which were the randomization process, deviation from the intended intervention, missing outcome data, measurement of the outcome, and selection of the reported result. The quality assessment results were classified to low risk, some problems, and high risk. Any study evaluated as high risk in any one of the five aspects were considered as high risk.

### 2.4. Subgroup Division

The subgroup was divided by the intensity level of the exercise. The included studies were classified according to the classifications of resistance training intensity provided by the American College of Sports Medicine (ACSM), which are medium-low intensity resistance training (20-less than 75% of 1RM; repetition maximum) and high intensity resistance training (75–100% of 1RM) [15]. The repetition of exercise was converted to intensity level according to the “Repetition Percentages of 1RM” developed by Michael Clark, and the converted intensity values (%; based on 1RM) were substituted into the corresponding study [16]. This method was also used in the classification of resistance training intensity in Liu’s study [10]. For progressive resistance training, the training intensity was taken as the middle value between the lowest and highest intensity value.

### 2.5. Data Analysis

The meta-analysis was performed using R Version 4.1.3, and the collated data were in the format of continuous data with the same units. The weighted mean difference (MD) and 95% confidence interval (CI) were used as effect size indicators, and the heterogeneity of the studies was evaluated by the *I*^2^ and *p*-value of the Q-test. If *I*^2^ ≥ 50% or *p* ≤ 0.1, the study was said to have heterogeneity among the included experiments. A random effects model was also used; if *I*^2^ ≥ 75%, the difference was significant, requiring careful consideration of whether to use meta-analysis. A fixed effects model combined the weighted mean differences for *I*^2^ ≤ 50, and a random effects model was used for *I*^2^ > 50%. The study conducted subgroup analyses for each of the included physiological indicators according to the intensity of resistance training. The differences between subgroups were analyzed using a random effects model combined values for independent sample *t*-test.

## 3. Results

### 3.1. Literature Screening and Quality Assessment

Searching for studies published up until 5 April 2022 by medicine subject heading terms and test words retrieved a total of 19,812 studies, 1359 of the screened studies related to randomized controlled trials, and 18 references were added from the published literature during manual screening for a total of 1377. A total of 1377 articles were manually screened, and 464 overlapped articles were excluded, leaving 873 articles. A total of 251 articles were screened out by reviewing the title and abstract, and 211 articles were excluded after the full texts were reviewed. In all, 41 articles included not only type 2 diabetic patients but also non-type 2 diabetic patients as subjects, 87 papers determined the control group to include non-stretching additional exercise, 69 papers had no experimental group accepted resistance training as interventions, the trial periods of 6 papers were too short, 3 papers were not written in English, and the data were unavailable for 5 papers. The final 40 publications were included in the meta-analysis for a literature quality assessment. The detailed process is shown in Figure 1.

The RoB2 tool was used to assess the quality of the included studies, and four high-risk papers were excluded. The research of Johannsen et al. [17] was excluded because of missing experimental results and the high risk of selective publication. Studies by Plotnikoff et al. [18] and Mavros et al. [19] were excluded due to the high risk in terms of result data loss. Church et al. [20] were excluded due to the high risk of randomization. The details of the excluded literature from the literature quality assessment are shown in Figure 2.

A total of 36 publications were included after quality assessment, and the specific regions with type 2 diabetic subjects included in the literature numbered 1491. Classified by region, the number of studies and subjects were: Asia (19, 540), Europe (5, 170), North America (6, 618), Oceania (3, 91), and South America (3, 72). Four studies included only male subjects, four studies included only female subjects, and four studies did not give a specific gender ratio. The other 24 studies included both males and females, with ages ranging from 19–73 years, pre-experimental glycosylated hemoglobin values ranging from 6.75% up and down to 9.51% up and down, and the duration of intervention for different groups ranging from 6–52 weeks, with specific information shown in Table 1.

### 3.2. Effect of Different Intensities of Resistance Training on the Adjunctive Therapy of Type 2 Diabetes Patients

The therapeutic effects of different intensities of resistance training on blood glucose, blood lipids, blood pressure, cardiopulmonary function, and anthropometrical indices in type 2 diabetic patients are shown in Table 2. Appendix A shows the screenshots of subgroup analysis for meta-analysis (source of data in Table 2).

#### 3.2.1. Blood Glucose Indicators

The studies on HbA1c (%) included 30 randomized controlled trials, with 377 patients in the high intensity experimental group and 377 patients in the control group and 239 in the medium-low intensity experimental group and 247 in the control group. The results showed that resistance training had a positive adjunctive therapeutic effect on HbA1c in type 2 diabetic patients (MD = −0.41, 95% CI: [−0.64, −0.18], *I*^2^ = 67%, Test for overall effect *p* < 0.01), and both high intensity resistance training and medium-low intensity resistance training had a positive adjunctive therapeutic effect on HbA1c, with the treatment effect being marginally better in the case of high intensity compared to medium-low intensity resistance training (MD = −0.49, 95% CI: [−0.73, −0.02], *I*^2^ = 75%, Test for overall effect *p* = 0.02; MD = −0.33, 95% CI: [−0.75, −0.13], *I*^2^ = 56%, Test for overall effect *p* = 0.01), but the difference was not statistically significant (*p* = 0.5268).

The studies on insulin (ng/mL) included 12 randomized controlled trials, with 60 patients in the high intensity experimental group and 59 in the control group and 120 in the medium-low intensity experimental group and 126 in the control group. The results showed that resistance training reduced the insulin value of type 2 diabetic patients. However, the difference between the experimental and control groups was not statistically significant (MD = −1.27, 95% CI: [−2.79, 0.26], *I*^2^ = 77%, Test for overall effect *p* = 0.10). Insulin values in type 2 diabetic patients were lower in the experimental group than in the control group in both subgroups of outcome indicators after experiencing high intensity and low-medium-intensity resistance training, but this difference was not significant (MD = −1.05, 95% CI: [−3.14, 0.55], *I*^2^ = 0%, Test for overall effect *p* = 0.35; MD = −1.30, 95% CI: [−3.14, 0.55], *I*^2^ = 83%, Test for overall effect *p* = 0.17). The difference between the two subgroups was also not significant (*p* = 0.868).

The studies on HOMA-IR included 10 randomized controlled trials, with 55 patients in the high intensity experimental group and 55 in the control group and 100 participants in the medium-low intensity experimental group and 99 in the control group. Resistance training had a significant adjunctive therapeutic effect on reducing HOMA-IR in type 2 diabetic patients (MD = −0.82, 95% CI: [−1.46, −0.18], *I*^2^ = 85%, Test for overall effect *p* = 0.01) but only in the high intensity subgroup, with the mean of inter-group difference of the experimental group being higher than the control group. However, this difference was not significant. The mean of inter-group differences in the medium-low intensity subgroup of the experimental group was lower than the control group, and this difference was statistically significant (MD = −0.82, 95% CI: [−1.46, −0.18], *I*^2^ = 0%, Test for overall effect *p* = 0.66; MD = −1.09, 95% CI: [−1.83, −0.36], *I*^2^ = 87%, Test for overall effect *p* < 0.01). There was a significant difference between these two subgroups (*p* = 0.0085).

The studies on fasting blood glucose (mmol/L) included 21 randomized controlled trials, with 119 patients in the high intensity experimental group and 121 in the control group and 188 in the medium-low intensity experimental group and 196 in the control group. Resistance training had an adjunctive therapeutic effect on reducing fasting blood glucose value in type 2 diabetic patients. After resistance training, the mean difference between fasting blood glucose in the experimental and control groups was significant (MD = −0.52, 95% CI: [−1.00, −0.04], *I*^2^ = 71%, Test for overall effect *p* = 0.03). High intensity resistance training was not as effective as medium-low intensity training, and only the mean difference between the experimental and control groups within the medium-low intensity subgroup was significant (MD = −0.23, 95% CI: [−1.37, 0.91], *I*^2^ = 76%, Test for overall effect *p* = 0.69; MD = −0.66, 95% CI: [−1.18, −0.15], *I*^2^ = 66%, Test for overall effect *p* = 0.01). The mean difference between the high intensity and medium-low intensity subgroups did not differ between groups (*p* = 0.4958).

#### 3.2.2. Blood Lipid Indicators

To understand the adjunctive therapeutic effect of triglycerides (mmol/L), 19 randomized controlled trials were included, with 227 patients in the high intensity experimental group and 228 in the control group and 152 in the medium-low intensity experimental group and 162 in the control group. After a certain period of resistance training, patients in the experimental group had significantly lower triglyceride levels (MD = −0.20, 95% CI: [−0.32, −0.08], *I*^2^ = 30%, Test for overall effect *p* < 0.01). The treatment effect was better in the high intensity subgroup than in the medium-low intensity subgroups, and only the mean values of the experimental and control groups within the high intensity subgroup were significantly different (MD = −0.28, 95% CI: [−0.44, −0.12], *I*^2^ = 0%, Test for overall effect *p* < 0.01; MD = −0.06, 95% CI: [−0.35, 0.22], *I*^2^ = 55%, Test for overall effect *p* = 0.67). Furthermore, the difference between the means of the high intensity and medium-low intensity subgroups was not significant between groups (*p* = 0.1917).

In terms of the adjunctive therapeutic effect of total cholesterol (mmol/L), 17 randomized controlled trials were included, including 108 patients in the high intensity experimental group and 107 in the control group and 166 in the medium-low intensity experimental group and 179 in the control group. The total cholesterol index of the experimental group was significantly lower than that of the control group after a period of resistance training (MD = −0.26, 95% CI: [−0.42, −0.09], *I*^2^ = 0%, Test for overall effect *p* < 0.01). The treatment effect was better in the high intensity group than in the medium-low intensity group, and the difference between the means of the experimental and control groups was significant in the high intensity group and not significant in the medium-low intensity group (MD = −0.32, 95% CI: [−0.56, −0.08], *I*^2^ = 22%, Test for overall effect *p* < 0.01; MD = −0.20, 95% CI: [−0.43, 0.03], *I*^2^ = 0%, Test for overall effect *p* = 0.09). Furthermore, the difference in the means between the high intensity and medium-low intensity subgroups was not significant between groups (*p* = 0.5884).

In the study of HDL cholesterol (mmol/L), 20 randomized controlled trials were included, with 240 patients in the high intensity experimental group and 241 in the control group and 166 in the medium-low intensity experimental group and 179 in the control group. After a period of resistance training, the mean HDL cholesterol levels were lower in the experimental group than in the control group (MD = −0.02. 95% CI: [−0.06, 0.02], *I*^2^ = 10%), which went against the expectation of researchers. However, this difference was not significant (*p* = 0.38). The mean value of the HDL cholesterol experimental group results was slightly lower in the high intensity group than in the control group. The difference between the experimental and control data was not significant (MD = −0.01, 95% CI: [−0.06, 0.05], *I*^2^ = 34%, Test for overall effect *p* = 0.81), and the HDL cholesterol experimental group results in the medium-low intensity groups were lower than those of the control group, but the difference was also not significant (MD = −0.03, 95% CI: [−0.09, 0.03], *I*^2^ = 0%, Test for overall effect *p* = 0.30). The difference in means between the high intensity and medium-low intensity subgroups was insignificant between groups (*p* = 0.5243).

In the study of LDL cholesterol (mmol/L), 18 randomized controlled trials were included, including 206 patients in the high intensity experimental group and 206 in the control group and 180 in the medium-low intensity experimental group and 193 in the control group. After a period of resistance training, the results were significantly lower in the experimental group than in the control group (MD = −0.18, 95% CI: [−0.30, −0.05], *I*^2^ = 0%, Test for overall effect *p* < 0.01). The treatment effect was better in the high intensity group than in the medium-low intensity group. There was a significant difference between the mean values of the experimental and control groups in the high intensity group and a difference which was not significant between the mean values of the experimental and control groups in the medium-low intensity group (MD = −0.19, 95% CI: [−0.35, −0.03], *I*^2^ = 6%, Test for overall effect *p* = 0.02; MD = −0.16, 95% CI: [−0.34, 0.03], *I*^2^ = 0%, Test for overall effect *p* = 0.10). The mean difference between the high intensity and medium-low intensity subgroups was not significantly different between groups (*p* = 0.8697).

#### 3.2.3. Blood Pressure

A total of 17 randomized controlled trials were included in terms of the study of diastolic blood pressure (mmHg), with 161 patients in the high intensity experimental group and 165 in the control group and 159 in the medium-low intensity experimental group and 166 in the control group. After a period of resistance training, the mean blood pressure in the experimental group was lower than the mean blood pressure in the control group, but this difference was not significant (MD = −1.81, 95% CI: [−4.80, 1.19], *I*^2^ = 81%, Test for overall effect *p* = 0.24). The difference between experimental and control group means was slightly greater in the high intensity subgroup than in the medium-low intensity subgroup. However, the differences between the experimental and control groups within both subgroups were not significant (MD = −2.16, 95% CI: [−5.99, 1.66], *I*^2^ = 81%, Test for overall effect *p* = 0.27; MD = −1.33, 95% CI: [−6.39, 3.74], *I*^2^ = 83%, Test for overall effect *p* = 0.61). Furthermore, the difference in means between the high intensity and medium-low intensity subgroups was not significant between groups (*p* = 0.7956).

A total of 17 randomized controlled trials were included in the study of systolic blood pressure (mmHg), including 161 patients in the high intensity experimental group and 165 in the control group and 159 in the medium-low intensity experimental group and 166 in the control group. After a period of resistance training, the mean systolic blood pressure in the experimental group was significantly lower than the mean value in the control group (MD = −6.83, 95% CI: [−11.50, −2.61], *I*^2^ = 72%, Test for overall effect *p* < 0.01). The reduction in the mean of systolic blood pressure in the high intensity group was not as great as that in the medium-low intensity group. Only within the medium-low intensity subgroup did the experimental results differ significantly between the experimental and control groups (MD = −4.36, 95% CI: [−9.74, 1.02], *I*^2^ = 71%, Test for overall effect *p* = 0.11; MD = −9.53, 95% CI: [- 16.15, −2.91], *I*^2^ = 71%, Test for overall effect *p* < 0.01). Furthermore, the reduction in inter-group means in both groups was not significantly different between groups (*p* = 0.2350).

#### 3.2.4. Cardiopulmonary Function Indicators

A total of six randomized controlled trials were included in the study of the resting heart rate (bpm), including 33 patients in the high intensity experimental group and 35 in the control group and 55 in the medium-low intensity experimental group and 56 in the control group. After a period of resistance training, the resting heart rate was significantly lower in the experimental group than in the control group (MD = −3.42, 95% CI: [−8.92, 2.09], *I*^2^ = 71%, Test for overall effect *p* = 0.22). The mean difference in resting heart rate between the experimental and control groups was slightly higher in the high intensity subgroup than in the medium-low intensity subgroup. The inter-group differences were not significant in both subgroups. (MD = −3.42, 95% CI: [−8.92, 2.09], *I*^2^ = 87%, Test for overall effect *p* = 0.43; MD = −2.71, 95% CI: [−8.05, 2.63], *I*^2^ = 0%, Test for overall effect *p* = 0.32). The mean difference between the high intensity and medium-low intensity subgroups was not significantly different (*p* = 0.8661).

A total of two randomized controlled trials were included in the study of maximal heart rate (bpm). The experimental group intervention was all high intensity resistance training, with 79 patients in the experimental group and 76 in the control group. After a period of high intensity resistance training, the mean maximal heart rate in the experimental group was lower than that in the control group, but the difference was not significant (MD = −0.10, 95% CI: [−0.83, 0.63], *I*^2^ = 0%. Test for overall effect *p* = 0.80).

A total of 12 randomized controlled trials were included in the study of maximal oxygen uptake (ml/kg), with 121 patients in the high intensity experimental group and 116 in the control group and 104 in the medium-low intensity experimental group and 105 in the control group. After a period of resistance training, the mean of maximal oxygen uptake was higher in the experimental group than in the control group, but the difference was not significant (MD = 0.62, 95% CI: [−0.99, 2.23], *I*^2^ = 81%, Test for overall effect *p* = 0.45). The mean of maximum oxygen uptake in the experimental group of high intensity was lower than that of the control group, but the difference was not significant (MD = −0.25, 95% CI: [−3.00, 2.49], *I*^2^ = 69%, Test for overall effect *p* = 0.86). The mean of maximum oxygen uptake in the experimental group of medium and low intensity was higher than that of the control group but was not significant (MD = 1.13, 95% CI: [−0.77, 3.0], *I*^2^ = 83%, Test for overall effect *p* = 0.24), and the mean difference between the high intensity subgroup and the medium-low intensity subgroup was not significant between groups (*p* = 0.4154).

#### 3.2.5. Anthropometrical Indicators

A total of 19 randomized controlled trials were included in the body weight study, with 195 patients in the high intensity experimental group and 196 in the control group and 173 in the medium-low intensity experimental group and 184 in the control group. After a period of resistance training, the mean weight of the experimental group was higher than that of the control group, with no significant difference (MD = 0.39, 95% CI: [−1.19, 1.97], *I*^2^ = 0%, Test for overall effect *p* = 0.63). Only the mean value of the experimental group was lower than that of the control group within the medium-low intensity subgroup groups. Only the experimental and control groups were statistically different within the high intensity group (MD = 4.25, 95% CI: [0.27, 8.22], *I*^2^ = 0%, Test for overall effect *p* = 0.04; MD = −0.33, 95% CI: [−2.05, 1.39], *I*^2^ = 0%, Test for overall effect *p* = 0.70). The difference between the high intensity and low-medium-intensity subgroups was significant in terms of the effect on body weight in patients with type 2 diabetes (*p* = 0.0382).

A total of 10 randomized controlled trials were included in the study of waist circumference (cm), including 100 patients in the high intensity experimental group and 103 in the control group and 102 in the medium-low intensity experimental group and 105 in the control group. After a period of resistance training, the mean reduction in waist circumference was greater in the experimental group than in the control group, with a non-significant difference (MD = −0.75, 95% CI: [−2.24, 0.75], *I*^2^ = 0%, Test for overall effect *p* = 0.33). The within-group difference was greater in the high intensity subgroup than in the medium-low intensity subgroup, and this within-group difference was not statistically significant (MD = −1.49, 95% CI: [−5.55, 2.58], *I*^2^ = 0%, Test for overall effect *p* = 0.47; MD = −0.63, 95% CI: [−2.24, 0.98], *I*^2^ = 0%, Test for (Test for overall effect *p* = 0.44). The difference in means between the high intensity and medium-low intensity subgroups was not significantly different between the two groups (*p* = 0.7006).

A total of 9 randomized controlled trials were included in the study of waist-to-hip ratio, including 12 patients in the experimental group and 12 in the control group in the high intensity group and 102 in the experimental group and 103 in the control group in the medium-low intensity group. After a period of resistance training, the mean waist circumference in the experimental group was smaller than the mean waist circumference in the control group, and the difference was not significant (MD = −0.02, 95% CI: [−0.03, 0.01], *I*^2^ = 0.2%, Test for overall effect *p* = 0.21). The waist-to-hip ratio was higher in the experimental group with high intensity than in the control group and lower in the experimental group with medium-low intensity than in the control group, with the difference not being significant in either group (MD = 0.01, 95% CI: [−0.06, 0.08]; MD = −0.02, 95% CI: [−0.04, 0.01], *I*^2^ = 5%, Test for overall effect *p* = 0.14). There was no significant difference between groups in the mean difference between the high intensity subgroup and the medium-low intensity subgroup (*p* = 0.4213).

A total of 6 randomized controlled trials were included in the study of fat mass (FM, kg), including 115 patients in the experimental group and 119 in the control group in high intensity group and 32 in the experimental group and 33 in the control group in the medium-low intensity group. After a period of resistance training, the mean fat mass value in the experimental group was lower than in the control group, and the difference was not significant (MD = −1.18, 95% CI: [−3.75, 1.39], *I*^2^ = 51.1%, Test for overall effect *p* = 0.37). The value of the difference between the experimental and control groups was greater in the high intensity group than in the medium-low intensity groups, and only the difference between the experimental and control groups in the medium-low intensity groups was significant (MD = −2.12, 95% CI: [−7.25, 3.01], *I*^2^ = 64%, Test for overall effect *p* = 0.42; MD = −1.57, 95% CI: [−3.03, −0.11], *I*^2^ = 43%, Test for overall effect *p* = 0.03). The mean difference between the high intensity and medium-low intensity subgroups was not significantly different (*p* = 0.7281).

A total of 13 randomized controlled trials were included in the study of the percentage of fat mass (PBF, %), with 103 patients in the experimental group and 108 in the control group in high intensity group and 140 in the experimental group and 143 in the control group in the medium-low intensity group. After a period of resistance training, the mean of the experimental group was lower than the that of the control group, and this difference was significant (MD = −1.38, 95% CI: [−2.64, −0.12], *I*^2^ = 52.4%, Test for overall effect *p* = 0.03). The difference between the experimental and control groups was greater for the high intensity groups than for the medium-low intensity groups. Only the difference between the experimental and control groups was significant for the medium-low intensity groups (MD = −2.69, 95% CI: [−7.98, 2.75], *I*^2^ = 76%, Test for overall effect *p* = 0.34; MD = −0.94, 95% CI: [−1.52, −0.36], *I*^2^ = 46%, Test for overall effect *p* < 0.01). The difference in means between the high intensity and medium-low intensity subgroups was insignificant between groups (*p* = 0.6455).

A total of 22 randomized controlled trials were included in the study of BMI, with 216 patients in the experimental group and 218 in the control group in the high intensity group and 193 in the experimental group and 192 in control the medium-low intensity group. After a period of resistance training, the mean of the experimental group was lower than that of the control group, and the difference was not significant (MD = −0.29, 95% CI: [−0.70, 0.11], *I*^2^ = 0%, Test for overall effect *p* = 0.15). The mean of the experimental group in the high intensity group was higher than the control group. The mean of the experimental group in the medium-low intensity group was lower than the control group, and none of the differences was significant (MD = 0.18, 95% CI: [−1.05,1.41], *I*^2^ = 0%, Test for overall effect *p* = 0.77; MD = −0.35, 95% CI: [−0.78, 0.08], *I*^2^ = 0%, Test for overall effect *p* = 0.11). The mean difference between the experimental and control groups was not significantly different between the high intensity and medium-low intensity subgroups (*p* = 0.4223).

#### 3.2.6. Adverse Events

Adverse events were pooled for all studies included in the meta-analysis (including high risk studies), and only a small number of studies reported hypoglycemic and other events. The Yavari et al. [38] study reported that two patients withdrew from the trial in the first month with recurrent hypoglycemia, Castaneda et al. [53] reported seven hypoglycemic events in the control group, and Church et al. [20] reported 8 serious adverse events in the resistance training group. However, no events were related to the intervention, including diverticulitis, emergency hysterectomy, lung cancer, and 5 cardiovascular diseases; only one adverse event was related to exercise, which was not specifically reported. Jorge et al. [39] reported a similar incidence of hypoglycemic events in the control and exercise groups. The Oliveira et al. [35] study reported two hypoglycemic events, and Plotnikoff et al. [18] reported 8 cases of musculoskeletal muscle injury in the resistance training group, where skeletal muscle injury affected the participants’ ability to perform resistance training, but no serious adverse events occurred. Two trials by Reid et al. [57] and Sigal et al. [50] were discussed the same randomized controlled trial, which reported one adverse event in the control group. Yamamoto et al. [27] reported one case of hospitalization for depression in the control group, one case of surgery for lumbar stenosis, and one case of hospitalization for complete AV block.

## 4. Discussion

To address the effect of resistance training of different intensities on blood glucose levels in patients with type 2 diabetes, Liu et al. [10] conducted a meta-analyses on HbA1c, including 11 high intensity and 9 medium-low intensity trials, and a meta-analyses on insulin, which had 5 high intensity and 5 medium-low intensity trials, for a total of 24 randomized controlled trials, and concluded that high intensity resistance training had greater benefits relative to medium-low intensity in terms of HbA1c and insulin attenuation, with significant differences between groups. However, our meta-analysis on glycemic control and HbA1c included 15 high intensity randomized controlled trials and 15 medium-low intensity trials; and on insulin, which included 3 high intensity and 9 medium-low intensity trials. For a total of 31 trials, with the results showing that the differences in HbA1c and insulin attenuation between high intensity and medium-low intensity groups were not statistically significant.

Differing from the study of Liu et al. [10], this study discarded the Chinese Wanfang database they selected, and added two internationally recognized databases, Cochrane Library and WOS. In terms of subgroup classification, this study classified randomized controlled trials with more than half of the duration being high intensity (≥75% 1RM) as high intensity subgroups and the rest as medium-low intensity subgroups according to ACSM guidelines. However, Liu et al.’s study [10] gave no specific subgroup classification criteria. To sum up, our study was more recognized in terms of database selection, more comprehensive in terms of search strategy, and clearer in terms of subgroup delineation criteria. The study demonstrated again the role of resistance training in glycemic control in type 2 diabetic patients, with a more significant effect on HbA1c, insulin, and fasting glucose stabilization, with no significant difference between high intensity resistance training and medium-low intensity resistance training in these three aspects. In Liu et al.’s study [10], Mahdirejei et al. [33] and Kadoglou et al. [36] were classified in the high intensity subgroup; but, since the the duration of the experiment was less than half, these papers were classified in the medium-low intensity subgroup. In Liu et al.’s study [10], the studies by Ishii et al. [55] and Gordon et al. [58] did not find data related to HbA1c. Avros et al. [19], Church et al. [20], and Plotnikoff et al. [18] were excluded from this study because of their high risk. All other randomized controlled trials were included in our study, except those in the Wanfang database.

Holten et al. [59] proved that resistance training increases the protein content of GLUT4 (glucose transporter 4), insulin receptor, protein kinase B-α/β, glycogen synthase (GS), and GS total activity. This increase in glucose clearance efficiency caused by resistance training exceeded the effect of increased muscle mass. A prospective study from the UK on the risk of complications associated with diabetes and HbA1c levels concluded that a 1% decrease in HbA1c decreased the risk of complications by 21%, and in our study, the difference in HbA1c within the high intensity subgroup was 0.16% lower than the medium-low intensity subgroups, which leads to the assumption that using high intensity resistance training as an intervention would reduce 3.4% diabetes complications risk more than using medium-low intensity resistance training; and it is also superior to medium-low intensity training in terms of glycemic control. The study demonstrated for the first time by meta-analysis that high intensity and medium-low intensity resistance training had a better adjunctive therapeutic effect on HOMA-IR. This effect was mainly caused by medium-low intensity resistance training. There was a significant difference in the therapeutic effect between medium-low intensity and high intensity resistance training.

TG and LDL were also found to be more predictive of atherosclerotic cardiovascular disease than HbA1c in type 2 diabetic patients. However, no meta-analysis on the effect of resistance training on lipid levels in type 2 diabetes was found in the current research. This study is the first research validated by meta-analysis to show that resistance training is effective in reducing triglycerides, total cholesterol, and LDL cholesterol levels in type 2 diabetic patients, and high intensity resistance training causes a greater reduction in triglycerides, total cholesterol, and LDL cholesterol levels than medium-low intensity resistance training. High-density lipoprotein cholesterol is a protective factor for coronary heart disease, and resistance training does not significantly increase HDL cholesterol levels in the blood. However, relatively speaking, the reduction in HDL cholesterol concentration caused by high intensity resistance training is smaller than that caused by medium-low intensity resistance training [60]. Therefore, although there was no significant difference between high intensity and medium-low intensity resistance training in terms of blood lipids, the treatment effect was slightly better in the case of high intensity than medium-low intensity training.

This meta-analysis study proved that resistance training reduces diastolic and systolic blood pressure in patients with type 2 diabetes. Due to the complexity of the association between blood pressure and mortality, it would appear unsuitable to use it as a critical indicator for making medical decisions. However, a cohort study by Pastor-Barriuso [61] demonstrated that a reduction in systolic blood pressure reduced the risk of death from cardiovascular disease of elderly patients aged ≥ 65 years, with a u-shaped correlation between diastolic blood pressure and the risk of death from cardiovascular disease. In our study, all type 2 diabetes patients aged ≥ 17 were included, hence it could not prove that the reduction in systolic and diastolic blood pressure caused by high intensity and medium-low intensity resistance training can be considered as a critical indicator for clinicians to make treatment decision regarding intensity selection.

It is worth noting that this meta-analysis showed significant differences in the effects of the two subgroups (high intensity and medium-low intensity) on both the HOMA-IR and body weight. High intensity resistance training caused a slightly increase in HOMA-IR, which was not the effect the researchers expected. In addition, high intensity resistance training also caused an increase in body weight, while the corresponding FM (kg) and PFM (%) did not increase. This result arose as result of high intensity exercise being more effective in terms of FM increase than medium-low exercise, which explained the reason for the slightly increased BMI value.

Based on the information in the literature included in the meta-analysis, the use of resistance training as adjunctive therapy for type 2 diabetic patients has been recognized by researchers. However, there is still a lack of certainty surrounding the question of whether to use high or medium-low intensity exercise. In this research, there is generally no significant difference in the adjunctive therapeutic effectiveness between high intensity and medium-low intensity resistance training for type 2 diabetic patients. However, according to the differences in experiment results between the experimental and control groups, high intensity resistance training was found have a greater effect in terms of reducing the HbA1c, triglyceride, total cholesterol, and LDL cholesterol levels of type 2 diabetic patients than medium-low intensity resistance training, with it also having a greater effect on diastolic blood pressure, resting mental, waist circumference, FM (kg), and PFM (%).

However, medium-low intensity resistance training had a greater effect on reducing insulin, fasting glucose, HDL cholesterol levels, and a slightly greater effect on systolic blood pressure than high intensity training. In addition, the effect of high and medium-low intensity resistance training on HOMA-IR, maximal oxygen uptake, body weight, waist-to-hip ratio, and BMI were opposite. These differences were not statistically significant except in the cases of HOMA-IR and body weight.

The intervention group with high intensity resistance training had higher HOMA-IR, body weight, waist-to-hip ratio, and BMI values and lower maximal oxygen uptake than the control group, but the intervention group with medium-low intensity resistance training had lower HOMA-IR, body weight, waist-to-hip ratio, BMI values and higher maximal oxygen uptake than the control group.

Although this study could be the most comprehensive review by far, it is undeniable that: first, there was high heterogeneity in a portion of the meta-analysis literature, and this heterogeneity may suggest selecting resistance training in more intensity levels as an intervention for randomized controlled trials to explore the optimal intensity of resistance training; second, resistance training as an adjunctive therapy should be a long-term procedure, but the trial periods of only two of the included randomized controlled trials were longer than two years, demonstrating that this study cannot validate the adjunctive therapeutic effect of long-term resistance training for type 2 diabetes.

## 5. Conclusions

Resistance training is a treatment that is effective in patients with type 2 diabetes. Although the *p*-value reflecting the difference in treatment effect between medium-low and high intensity resistance training did not reach statistical significance, the practical importance of treatment differences cannot be ignored. The weighted mean difference between the experimental and control groups within the two subgroups indicated that high intensity resistance training was slightly more effective than medium-low intensity training. The therapeutic effectiveness in terms of HbA1c indicated that choosing high intensity resistance training reduced the risk of diabetes complications 3.4% more comparing to medium-low intensity resistance training.

## Figures and Tables

**Figure 1 healthcare-11-00440-f001:**
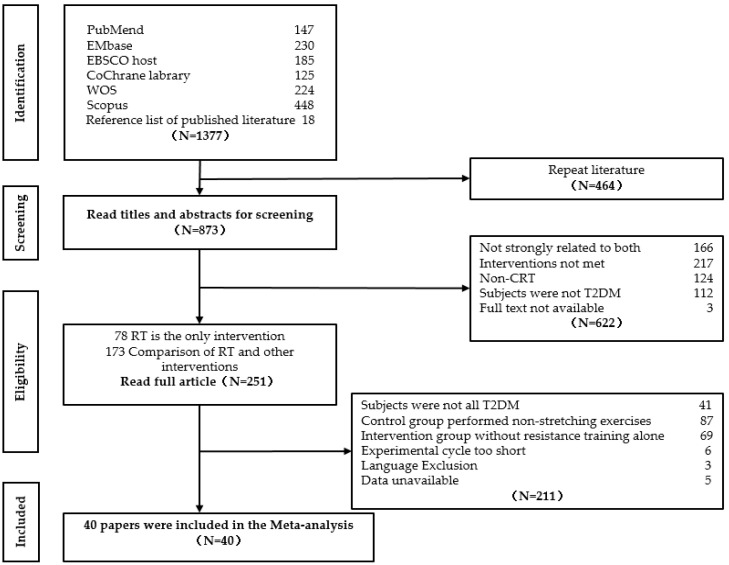
Flow chart of literature screening.

**Figure 2 healthcare-11-00440-f002:**
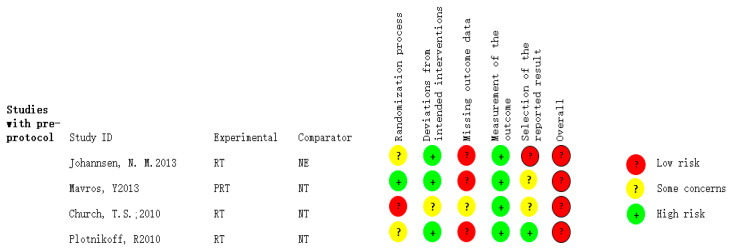
Specific information of the excluded literature.

**Table 1 healthcare-11-00440-t001:** Characteristics of the literature included in the meta-analysis.

Study	Country	RT /CN	Female %	Age (Year)	HbA1C (%)	Intensity %1RM	Repetition (Times)	Set	Frequency (t/wk)	Duration (k)	Outcome Indicators
Ramachandran et al. [21]	India	12/12	-	51.4 ± 2.2	8.34 ± 0.67	70–80↑	12	2	3	12	a.e.g.h.
Sabouri et al. [22]	Iran	15/13	53.6	51.76 ± 3.92	7.52 ± 0.88	80↑	8	3	3	12	n.s.m.k.i.j.d.b.c.a.e.f.g.h.
Ranasinghe et al. [23]	Sri Lanka	27/28	50	49.16 ± 8.13	8.27 ± 1.7	81↑	8	3	2	12	d.a.b.c.e.f.g.h.i.j.n.s.o.q.r.
Gholami et al. [24]	Iran	15/14	0	63.48 ± 3	9.51 ± 1.82	≥50↓	10–15	1–3	2–3	12	a.
Rezaeeshirazi et al. [25]	Iran	14/15	0	21.9 ± 1.97	≥2 years	50–70↓	8–15 ^ii^	3	4	8	d.b.c.m.s.r.q.o.n.
Motahari et al. [26]	Iran	15/13	-	44.33 ± 2.81	8.1 ± 0.8	40–80↓	8–18 ^ii^	3	3	12	n.s.m.r.c.a.
Yamamoto et al. [27]	Japan	18/17	45.7	73.25 ± 2.55	7.21 ± 0.81	Medium-low↓	20	15 Min	7	48	a.s.
Mogharnasi et al. [28]	Iran	10/8	100	48.52 ± 7.06	≥3 years	30–80↓	10–18 ^ii^	3	3	10	n.s.p.r.m.d.b.c.
Hsieh et al. [29]	China	14/15	63.3	71.2 ± 4.4	7.25 ± 0.76	40–75↓	8–12	3	3	12	m.l.i.j.n.r.o.d.a.e.f.g.h.
Botton et al. [30]	Brazil	13/13	40.9	69.6 ± 6.9	7.07 ± 0.67	67 ^i^↓	12	3	3	12	a.d.e.f.g.h.
AminiLari et al. [31]	Iran	12/15	100	45–60	≥2 years	50–55↓	8	3	3	12	n.s.r.d.b.
Shabani et al. [32]	Iran	10/10	100	50.75 ± 5.83	7.45 ± 1.43	40–65↓	8–12	1–3 ^iii^	3	8	n.s.p.a.d.
Mahdirejei et al. [33]	Iran	9/9	0	48.61 ± 7.88	7.96 ± 1.62	50–80↓	8–15 ^ii^	3	3	8	n.s.r.p.m.e.f.g.h.d.a.b.c
Kadoglou et al. [34]	Greece	23/24	70.2	57 ± 6.3	7.9 ± 0.75	60–80↓	8–10	2–3	4	24	s.p.i.j.m.d.a.e.f.g.h.b.c.r
Oliveira et al. [35]	Brazil	10/12	63.6	53.7 ± 9.4	7.71 ± 1.73	≥50↓	8–12	4	3	12	d.a.e.f.g.h.n.q.o.p.m.i.j.r.
Kadoglou et al. [36]	Greece	23/24	74.5	63.08 ± 4.87	7.45 ± 0.45	60–80↓	6–8	2–3	3	12	s.p.i.j.a.d.e.f.g.h.b.c.m.r.
Hameed et al. [37]	India	24/24	27.08	44.7 ± 4.9	8.4 ± 0.8	65–70↓	10	3	2–3	8	a.n.o.e.f.g.h.i.j.
Yavari et al. [38]	Iran	20/20	53.75	51.5 ± 7.48	7.5 ± 0.89	75–80↑	8–10	3	3	52	a.d.e.f.g.h.n.s.r.i.j.l.m.
Jorge et al. [39]	Brazil	12/12	62.5	53.8 ± 9.4	7.63 ± 1.79	89–94↑	10	2	3	12	s.p.m.i.j.a.d.f.g.c.e.
Larose et al. [40]	Canada	64/63	36.2	54.75 ± 58.6	7.69 ± 6.97	80 ^i^↑	8	2	2–3	22	a.n.s.m.
Kwon et al. [41]	Seoul, Korea	12/15	-	57.74 ± 5.88	7.23 ± 0.79	40–50↓	10–15 ^ii^	3	3	12	n.a.b.e.f.g.h.
Hazley et al. [42]	Britain	6/6	41.7	54 ± 9	7.3 ± 0.95	50–60↓	15	3	2	8	s.o.p.l.d.a.e.f.g.h.b.c.i.j.
Ku et al. [43]	Korea	13/16	100	56.86 ± 7.32	7.3 ± 0.8	40–50↓	15–20 ^ii^	3	5	12	n.s.o.a.d.
Gavin et.al. [44]	Canada	64/63	34.6	54.75 ± 7.35	7.69 ± 0.88	80 ^i^↑	8	2	2–3	22	e.a.g.h.
Wycherley et al. [45]	Australia	17/16	-	55.0 ± 8.4	7.45 ± 1.2	70–85↑	12	2	3	16	n.s.q.o.i.j.d.a.b.e.f.g.h.
Arora et al. [46]	India	9/10	50	54 ± 3.9	7.67 ± 1.18	70–80↑	10	3	2	8	a.g.e.f.i.j.l.s.
Cheung et al. [47]	Australia	20/17	67.6	60.38 ± 7.85	7.31 ± 1.36	67 ^i^↓	12	2	5	16	a.s.o.
Larose et al. [48]	Canada	64/63	36.2	54.75 ± 7.35	7.69 ± 0.88	80 ^i^↑	8	2	2–3	22	k.
Shenoy et al. [49]	India	10/10	40	54 ± 3.89	7.67 ± 0.41	60–100↑	10	3	2	16	a.d.i.j.l.
Sigal et al. [50]	Canada	56/59	37.5	54.75 ± 7.35	7.69 ± 0.88	81↑	8	2–3	3	22	a.i.j.e.f.g.h.n.o.q.r.s.
Baum et al. [51]	Germany	13/13	40	63.1 ± 6.64	-	70–80↑	12	3	3	36	d.
Brooks et al. [52]	USA	31/31	35.48	66 ± 1.58	8.55 ± 0.3	70–80↑	8	3	3	16	a.
Castaneda et al. [53]	USA	29/31	64.5	66 ± 1.58	8.55 ± 0.3	60–80↓	8	3	3	16	a.d.f.g.h.i.j.l
Dunstan et al. [54]	Australia	11/10	38	50.68 ± 6.79	4.8 years	50–55↓	10–15	2–3	3	8	n.s.p.b.d.i.j.l.a.
Ishii et al. [55]	Japan	9/8	0	49.2 ± 8.58	9.22 ± 2.5	40–50↓	10–20	2	5	6	s.r.
Honkola et al. [56]	Finland	18/20	55	64.63 ± 2	7.61 ± 1.31	medium↓	12–15	2	2	20	i.j.e.f.g.h.n.a.

“i” No specific exercise intensity is given in the text; intensity is defined by the number of single repetitions. “ii” The number of repetitions is the lowest number of repetitions—the highest number of repetitions in all phases. “iii” The number of sets is the lowest number of sets—the highest number of sets in all phases. “-” indicates that the corresponding value is not given in the literature and cannot be estimated. “↑” indicates that the intervention used high intensity resistance training or more than half high intensity training; “↓” indicates that the intervention used medium-low intensity resistance training. The duration of type 2 diabetes was used instead of glycosylated hemoglobin levels in the intervention and control groups where they were not given. Other ranges are taken as in the original text. a: HbA1c; b: insulin; c: HOMA-IR; d: fasting blood glucose; e: triglycerides; f: total cholesterol; g: high-density lipoprotein cholesterol; h: low-density lipoprotein cholesterol; i: diastolic blood pressure; j: systolic blood pressure; k: maximum heart rate; l: resting heart rate; m: maximum oxygen uptake; n: weight; o: waist circumference; p: waist-to-hip ratio; q: body Fat; r: percentage of body fat; s: BMI.

**Table 2 healthcare-11-00440-t002:** Effect of resistance training as an auxiliary to treatment in patients with type 2 diabetes.

Outcome	Participants RT/NT	Merger Effect Value MD 95%CI	Quantifying Heterogeneity		Test for Overall Effect	Between Groups
			*I*^2^ (%)	*p*	*p*	*p*
HbA1c (%) (30)	616/624	−0.41 [−0.64, −0.18]	67	<0.01	<0.01	
H	377/377	−0.49 [−0.73, −0.02]	75	<0.01	0.02	0.5268
L-M	239/247	−0.33 [−0.75, −0.13]	56	<0.01	0.01	
Insulin (ng/mL) (12)	180/185	−1.27 [−2.79, 0.26]	77	<0.01	0.10	
H	60/59	−1.05 [−3.26, 1.15]	0	0.68	0.35	0.8680
L-M	120/126	−1.30 [−3.14, 0.55]	83	<0.01	0.17	
HOMA-IR (10)	155/154	−0.82 [−1.46, −0.18]	85	<0.01	0.01	
H	55/55	0.11 [−0.40, −0.63]	0	0.85	0.66	0.0085
L-M	100/99	−1.09 [−1.83, −0.36]	87	<0.01	<0.01	
FBG (mmol/L) (21)	307/317	−0.52 [−1.00, −0.04]	71	0.053	0.03	
H	119/121	−0.23 [−1.37, 0.91]	76	<0.01	0.69	0.4958
L-M	188/196	−0.66 [−1.18, −0.15]	66	<0.01	0.01	
TG (mmol/L) (19)	379/390	−0.20 [−0.32, −0.08]	30	0.10	<0.01	
H	227/228	−0.28 [−0.44, −0.12]	0	0.88	<0.01	0.1917
L-M	152/162	−0.06 [−0.35, 0.22]	55	0.02	0.67	
TC (mmol/L) (17)	274/286	−0.26 [−0.42, −0.09]	0	0.49	<0.01	
H	108/107	−0.32 [−0.56, −0.08]	22	0.26	<0.01	0.5884
L-M	166/179	−0.20 [−0.43, 0.03]	0	0.62	0.09	
HDL-c (mmol/L) (20)	406/420	−0.02 [−0.06, 0.02]	10	0.33	0.38	
H	240/241	−0.01 [−0.06, 0.05]	34	0.14	0.81	0.5243
L-M	166/179	−0.03 [−0.09, 0.03]	0	0.63	0.30	
LDL-c (mmol/L) (18)	386/399	−0.18 [−0.30, −0.05]	0	0.64	<0.01	
H	206/206	−0.19 [−0.35, −0.03]	6	0.38	0.02	0.8697
L-M	180/193	−0.16 [−0.34, 0.03]	0	0.63	0.10	
DBP (mmHg) (17)	320/331	−1.81 [−4.80, 1.19]	81	<0.01	0.24	
H	161/165	−2.16 [−5.99, 1.66]	81	<0.01	0.27	0.7956
L-M	159/166	−1.33 [−6.39, 3.74]	83	<0.01	0.61	
SBP (mmHg) (17)	320/331	−6.83 [−11.50, −2.61]	72	<0.01	<0.01	
H	161/165	−4.36 [−9.74, 1.02]	71	<0.01	0.11	0.2350
L-M	159/166	−9.53 [−16.15, −2.91]	71	<0.01	<0.01	
Rest HR (bpm) (6)	88/91	−3.42 [−8.92, 2.09]	71	<0.01	0.22	
H	33/35	−3.42 [−8.92, 2.09]	87	<0.01	0.43	0.8661
L-M	55/56	−2.71 [−8.05, 2.63]	0	0.97	0.32	
HRmax (bpm) (2)	79/76	−0.10 [−0.83, 0.63]	0	0.81	0.80	
H	79/76	−0.10 [−0.83, 0.63]	0	0.81	0.80	
L-M						
VO2max (ml/kg) (12)	225/221	0.62 [−0.99, 2.23]	81	<0.01	0.45	
H	121/116	−0.25 [−3.00, 2.49]	69	0.01	0.86	0.4154
L-M	104/105	1.13 [−0.77, 3.04]	83	<0.01	0.24	
Weight (kg) (19)	368/380	0.39 [−1.19, 1.97]	0	0.64	0.63	
H	195/196	4.25 [0.27, 8.22]	0	0.75	0.04	0.0382
L-M	173/184	−0.33 [−2.05, 1.39]	0	0.76	0.70	
WC (cm) (10)	202/208	−0.75 [−2.24, 0.75]	0	0.85	0.33	
H	100/103	−1.49 [−5.55, 2.58]	0	0.82	0.47	0.7006
L-M	102/105	−0.63 [−2.24, 0.98]	0	0.64	0.44	
WHR (9)	114/115	−0.02 [−0.03, 0.01]	0	0.43	0.21	
H	12/12	0.01 [−0.06, 0.08]				0.4213
L-M	102/103	−0.02 [−0.04, 0.01]	5	0.39	0.14	
FM (kg) (6)	147/152	−1.18 [−3.75, 1.39]	51	0.07	0.37	
H	115/119	−2.12 [−7.25, 3.01]	64	0.04	0.42	0.7281
L-M	32/33	−1.57 [−3.03, −0.11]	43	0.18	0.03	
PFM (%) (13)	243/251	−1.38 [−2.64, −0.12]	52	0.01	0.03	
H	103/108	−2.61 [−7.98, 2.75]	76	0.02	0.34	0.6455
L-M	140/143	−0.94 [−1.52, −0.36]	46	0.05	<0.01	
BMI (22)	409/410	−0.29 [−0.70, 0.11]	0	0.92	0.15	
H	216/218	0.18 [−1.05, 1.41]	0	0.86	0.77	0.4223
L-M	193/192	−0.35 [−0.78, 0.08]	0	0.80	0.11	

HbA1c: glycosylated hemoglobin; HOMA-IR: homeostatic model assessment for insulin resistance; FBG: fasting blood glucose; TG: triglycerides; TC: total cholesterol; HDL-c: high-density lipoprotein cholesterol; LDL-c: low-density lipoprotein cholesterol; DBP: diastolic blood pressure; SBP: systolic blood pressure; HR: heart rate; VO2max: maximal oxygen consumption; WC: waist circumference; WHR: waist-to-hip ratio; FM: fat mass; PFM: percentage of fat mass; BMI: body mass index. The number in parentheses is the number of included studies.

## Data Availability

Data are contained within the article.

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
