# Peer review of "Intensity Differences of Resistance Training for Type 2 Diabetic Patients: A Systematic Review and Meta-Analysis"

_healthcare, 2023, doi:10.3390/healthcare11030440_

Round 1
Reviewer 1 Report
Comments:
1. In lines 16-17, “exercise intensity was categorized as low-and moderate intensity and high intensity”. In line 21, “therapeutic effects of high-intensity and medium-low intensity resistance training”. Please named the subgroup the same.
2. The author combines several groups of diabetic patients, how to make a general control? The difference in lifestyle, drug management, countries, gender, and age will contribute to the final results.
3. Pages 7-8, table 2, there is almost no significant difference between high-intensity and medium-low intensity resistance training groups except weight (kg). How to draw the conclusion that “the treatment effect of choosing high-intensity resistance training was slightly better than low-and-moderate intensity resistance training”? Could these be slight changes due to experiment errors?
4. In this case, I suggest the authors include no training diabetic groups as the baseline.
Author Response
Dear Respected Reviewer,
Thank you for giving us the opportunity to submit a revised draft of our manuscript titled " Intensity Differences of Resistance Training for Type 2 Diabetic Patients: A Systematic Review and Meta-Analysis '' (Manuscript ID: healthcare-2135867) to Nutrition and Public Health - Healthcare. We appreciate the time and effort that you and the reviewers have dedicated to providing your valuable feedback on our manuscript. We are grateful to the reviewers for their insightful comments on our paper. We strongly believe that the comments and suggestions have increased the scientific value of the revised manuscript by many folds.
We have been able to incorporate changes to reflect most of the suggestions provided by the reviewers. We hope the manuscript after careful revisions meet your high standards and with these changes and clarifications, our work might be suitable for publication in Nutrition and Public Health - Healthcare. The authors welcome further constructive comments if any. Please see below, in red, for a point-by-point response to the reviewers' comments. All modifications in the manuscript have been highlighted in yellow.
Reviewer Comments:
Reviewer 1:
Point-1: In lines 16-17, “exercise intensity was categorized as low-and moderate intensity and high intensity”. In line 21, “therapeutic effects of high-intensity and medium-low intensity resistance training”. Please named the subgroup the same.
Point-1 Authors’ response: Thank you for your suggestion. The "low-and moderate" has been revised as "medium-low".
Point-2: The author combines several groups of diabetic patients, how to make a general control? The difference in lifestyle, drug management, countries, gender, and age will contribute to the final results.
Point-2 Authors’ response: The quality assessment of previous studies research in the meta-analysis assessed the quality of the literature in five areas: randomization process, deviations from the intended intervention, missing outcome data, measurement of outcomes, and selection of reported outcomes. Previous studies eliminated high-risk studies that may affect the results of the meta-analysis. The quality assessment tool used in this study were RoB2 tools. The previous results of quality assessment have been listed in the attachment.
Ref:
Sterne, J. A., Savović, J., Page, M. J., Elbers, R. G., Blencowe, N. S., Boutron, I., ... & Higgins, J. P. (2019). RoB 2. BMJ: British Medical Journal, 366, 1-8.
Point-3: Pages 7-8, table 2, there is almost no significant difference between high-intensity and medium-low intensity resistance training groups except weight (kg). How to draw the conclusion that “the treatment effect of choosing high-intensity resistance training was slightly better than low-and-moderate intensity resistance training”? Could these be slight changes due to experiment errors?
Point-3 Authors’ response: No statistical difference, but its practical importance cannot be ignored (McShane et al. 2019). For example, in Table 2, the TG row, P=0.1917, indicates that there is a 19.17% chance that there is no difference in the treatment effect caused between the high intensity and low and medium intensity subgroups, and another 80.83% chance that there is a difference in the treatment effect between high intensity and low and medium intensity. And from the "Merger effect value", there is 80.83% possibility that the difference is better between high intensity and low and medium intensity. In the actual implementation process, it is better to choose the high intensity resistance training with "Merger effect value", and this study provides data to support this conclusion.
Ref:
McShane, B. B., Gal, D., Gelman, A., Robert, C., & Tackett, J. L. (2019). Abandon statistical significance. The American Statistician, 73(sup1), 235-245.
Point-4: In this case, I suggest the authors include no training diabetic groups as the baseline.
Point-4 Authors’ response: The control group of the study included in the Meta-analysis did not participate in resistance exercise or arranged only stretching (line96-98). Therefore, it is thought that there is no great difficulty in presenting the control group of this study as the baseline. If you have further comments on this part, we will revise it.

Reviewer 2 Report
The authors retrieved many papers, the manuscript data is rich, and the overall quality is high. However, the conclusion of the manuscript cannot highlight their findings, and some other issues need to be improved.
1. Line 96. why use "aged ≥17 years" as Inclusion Criteria?
2. Line 147-149, how screened 1319 literature go to "a total of 1377"?
3. Please check the data in Table 1. I have checked the data from the original paper written by Hameed et al. [37], their subjects' HbA1C for the resistance exercise group and control group are 8.68±0.9 and 8.29±0.7, but the author got HbA1C for all subjects are 8.4±1.6, How did they calculate the SD for all subjects?
4. Line 185, check the "HOMA-1R". And check Table 2 Outcome "HOMA-1R(10)".
5. Please provide the effect size of every outcome variable involved in Table 2 as the form of an attachment.
6. Table 2 needs one line between BMI (22) and H.
Author Response
Dear Respected Reviewer,
Thank you for giving us the opportunity to submit a revised draft of our manuscript titled " Intensity Differences of Resistance Training for Type 2 Diabetic Patients: A Systematic Review and Meta-Analysis '' (Manuscript ID: healthcare-2135867) to Nutrition and Public Health - Healthcare. We appreciate the time and effort that you and the reviewers have dedicated to providing your valuable feedback on our manuscript. We are grateful to the reviewers for their insightful comments on our paper. We strongly believe that the comments and suggestions have increased the scientific value of the revised manuscript by many folds.
We have been able to incorporate changes to reflect most of the suggestions provided by the reviewers. We hope the manuscript after careful revisions meet your high standards and with these changes and clarifications, our work might be suitable for publication in Nutrition and Public Health - Healthcare. The authors welcome further constructive comments if any. Please see below, in red, for a point-by-point response to the reviewers' comments. All modifications in the manuscript have been highlighted in yellow.
Reviewer Comments:
Reviewer 2:
The authors retrieved many papers, the manuscript data is rich, and the overall quality is high. However, the conclusion of the manuscript cannot highlight their findings, and some other issues need to be improved.
Point-1: Line 96. why use "aged ≥17 years" as Inclusion Criteria?
Point-1 Authors’ response: Thank you for your comments. In Yang et. al (2014), a comprehensive review of the effect of resistance exercise and aerobic exercise on the treatment of T2DM was conducted, and the screening criteria in Yang's review was "the participants were people with type 2 diabetes aged 18 years or more". In Yang's review, the screening criterion was "the participants were people with type 2 diabetes aged 18 years or more", and in order to echo Yang et. al's (2014) study, which further investigated the effect of different intensity of resistance exercise on the treatment effect of T2DM, the screening criterion also included "aged ≥17".
Ref:
Yang, Z., Scott, C. A., Mao, C., Tang, J., & Farmer, A. J. (2014). Resistance exercise versus aerobic exercise for type 2 diabetes: a systematic review and meta-analysis. Sports medicine, 44(4), 487-499.
Point-2: Line 147-149, how screened 1319 literature go to "a total of 1377"?
Point-2 Authors’ response: Thank you for the kindly checking. The number should be 1359, which has been corrected.
Point-3: Please check the data in Table 1. I have checked the data from the original paper written by Hameed et al. [37], their subjects' HbA1C for the resistance exercise group and control group are 8.68±0.9 and 8.29±0.7, but the author got HbA1C for all subjects are 8.4±1.6, How did they calculate the SD for all subjects?
Point-3 Authors’ response: Thank you for the corrections, the synthetic standard deviation was calculated according to the following formula.
(1)
The formula was derived from previous research named "Effect of benzene exposure on blood routine indexes: A meta-analysis". However, in the search for relevant English literature, it was found that the standard deviations were combined according to a different formula in the English literature.
(2)
From “Combining follow-up and change data is valid in meta-analyses of continuous outcomes: a meta-epidemiological study”
(3)
From:” Imputing missing standard deviations in meta-analyses can provide accurate results”
Equation (2) and equation (3) are the same, but not the same as equation (1) used in this study. The two equation were verified again, and it was found that the combined standard deviation estimated by equations (2) and (3) was closer to the actual standard deviation. Therefore, the combined standard deviations in the data of the age and HbA1c columns in Table 1 were recalculated according to equation (2). Many thanks to the reviewers for their help in finding this problem.
Point-4: Line 185, check the "HOMA-1R". And check Table 2 Outcome "HOMA-1R(10)".
Point-4 Authors’ response: The value has been corrected.
Point-5: Please provide the effect size of every outcome variable involved in Table 2 as the form of an attachment.
Point-5 Authors’ response: All data in Table 2 are derived from the forest maps corresponding to the indicators, and all forest maps have been provided in the form of annexes. We have attached the zip file you pointed out.
Point-6: Table 2 needs one line between BMI (22) and H.
Point-6 Authors’ response: Table 2 has been corrected.
Round 2
Reviewer 1 Report
Agree to accept.
Reviewer 2 Report
Thanks to the authors for considering the reviewers' comments and making point-by-point responses to the reviewer.